# Effect of MiRNA 204-5P Mimics and Lipopolysaccharide-Induced Inflammation on Transcription Factor Levels, Cell Maintenance, and Retinoic Acid Signaling in Primary Limbal Epithelial Cells

**DOI:** 10.3390/ijms26083809

**Published:** 2025-04-17

**Authors:** Maryam Amini, Tanja Stachon, Shao-Lun Hsu, Zhen Li, Ning Chai, Fabian N. Fries, Berthold Seitz, Swarnali Kundu, Shweta Suiwal, Nóra Szentmáry

**Affiliations:** 1Dr. Rolf M. Schwiete Center for Limbal Stem Cell and Congenital Aniridia Research, Saarland University, 66424 Homburg/Saar, Germany; maryam@amini.cloud (M.A.); tanja.stachon@uni-saarland.de (T.S.); arinahsu@gmail.com (S.-L.H.); zhen.li@uks.eu (Z.L.); ning.chai@uks.eu (N.C.); fabian.fries@uks.eu (F.N.F.); theswarnalikundu@gmail.com (S.K.); shweta.suiwal@uni-saarland.de (S.S.); 2Department of Ophthalmology, Saarland University Medical Center, 66424 Homburg/Saar, Germany; berthold.seitz@uks.eu

**Keywords:** miR-204-5p, limbal epithelial cells, lipopolysaccharide, inflammation, miRNA mimic

## Abstract

MicroRNA-204-5p (miR-204-5p) is a critical regulator of differentiation, structural maintenance, and inflammation in limbal epithelial cells (LECs). This study examined the role of miR-204-5p in modulating the gene expression related to transcription factors, cell structure, extracellular matrix remodeling, and retinoic acid signaling under normal and lipopolysaccharide (LPS)-induced inflammatory conditions. Using qPCR, we analyzed the mRNA levels of FOSL2, FOXC1, Meis2, PPARγ, ABCG2, PTGES2, IL-1β, IL-6, KRT3, KRT12, MMP2, MMP9, RARA, RARB, RXRA, RXRB, CRABP2, RBP1, RDH10, ADH7, ADH1A1, FABP5, CYP1B1, and CYP26A1, while changes in protein levels were assessed via Western blot or ELISA. Our data revealed that the overexpression of miR-204-5p reduced the mRNA levels of FOXC1, KRT12, and RDH10 under normal and inflammatory conditions (*p* ≤ 0.039). Additionally, it decreased FOSL2 and RXRA mRNA under normal conditions (*p* = 0.006, *p* = 0.011) and KRT3 and FABP5 mRNA under inflammatory conditions (*p* = 0.010, *p* = 0.001). The IL-6 mRNA expression was significantly increased following the LPS treatment in cells overexpressing miR-204-5p (*p* = 0.029). A protein analysis revealed significant reductions in FOXC1 and KRT3 in the miR-204-5p-transfected cells during LPS-induced inflammation (*p* = 0.020, *p* = 0.030). These findings suggest that miR-204-5p modulates genes critical to the differentiation, migration, and inflammatory response of LECs. The modulation of FOXC1 and KRT3 by miR-204-5p highlights these proteins as novel targets under inflammatory conditions.

## 1. Introduction

Micro RNAs (miRNAs) are non-coding single-stranded small (18–25 nt) RNAs that play a crucial role in gene regulation [1]. These miRNAs inhibit the expression of target mRNAs by binding to the 3’ untranslated regions (3’UTRs) and forming an RNA-induced silencing complex [2]. miR204-5p originates from the sixth intron of Transient Receptor Potential Melastatin 3 (TRPM3) and has been identified as a key player in various biological processes, including eye development [3,4], osteogenesis [5], and lipogenesis [6]. miR204-5p knockdown in the lens and retina of Medaka fish (*Oryzias latipes*) has been shown to result in lens abnormalities and eye coloboma. In this context, miR-204-5p has been demonstrated to regulate gene expression and the PAX6 pathway by targeting the transcription factor Meis2 [3]. Additionally, another study found that during lens development, several targets of miR204-5p were significantly upregulated in Pax6 mutant mice [4]. This observation suggests the potential regulatory effect of PAX6 on miR204-5p in these cells.

Moreover, miR204-5p downregulation has been observed during the wound-healing process in human corneal epithelial cells. Cells transfected with miR204-5p showed a significant reduction in their proliferation and migration [7]. Another study on corneal neovascularization demonstrated that the overexpression of miR-204 in injured corneas targets multiple genes, effectively inhibiting corneal vascularization [8]. A more recent study revealed that the overexpression of miR-204-5p suppressed the Angiopoietin 1 (ANGPT1) expression in both a human limbal epithelial cell line and primary limbal epithelial cells (LECs), thereby reducing the vascularization under normal conditions and during lipopolysaccharide (LPS)-induced inflammation [9]. In primary human LECs, miR204-5p overexpression led to the downregulation of ANGPT1 at both the mRNA and protein levels and under both normal and LPS-induced inflammatory conditions. Furthermore, a significant reduction in mRNA levels of Vascular Endothelial Growth Factor α (VEGFα) was shown during inflammation [9]. These findings underscore the critical role of miR204-5p and its expression levels in maintaining corneal health.

Congenital aniridia is a rare genetic disorder, primarily caused by PAX6 haploinsufficiency [10]. The main cause of progressive visual loss in congenital aniridia is so-called aniridia-associated keratopathy, or AAK. In this inflammatory condition, progressive limbal stem cell deficiency leads to corneal neovascularization from the periphery, resulting in the formation of a vascularized corneal pannus [9]. In conjunctival impression cytology samples from individuals with congenital aniridia, 26.8-fold downregulation in the expression of miRNA 204-5p has been observed [11]. These changes in conjunctival surface protein expression may influence the gene expression in neighboring LECs and differentiated corneal epithelial cells. In addition, in limbal epithelial cells with AAK, gene expression changes regarding transcription factors, cell maintenance, and retinoic acid signaling components have been described previously [12].

In our present study, we aimed to investigate the transcription factor levels, cell maintenance, and retinoic acid signaling in primary human LECs following exogenous miR-204-5p expression under both healthy and LPS-induced inflammatory conditions.

## 2. Results

Our manuscript includes a secondary analysis following the use of microRNA-204-5p mimics and LPS treatment in LECs. The transfection of the miRNA 204-5p mimics was successfully performed in our LECs, as described by Abbasi et al. [9].

### 2.1. The Effect of miR204-5p Mimics on Cell Proliferation, mRNA, and Protein Levels of Transcription Factors

To compare the effects of miR-204-5p overexpression under normal and inflammatory conditions, a BrdU assay was performed. Our results demonstrated a significant reduction in the proliferation rate of the cells transfected with miR-204-5p, both in the absence and presence of the LPS treatment. In contrast, the LPS treatment alone did not affect the proliferation of the LECs (Figure 1).

Our previous findings demonstrated that the exogenous expression of miR-204-5p has only a minor impact on the PAX6 expression in LECs [9]. This led us to investigate whether miR-204-5p modulates the transcription in LECs by targeting another transcription factor. To address this, we analyzed the gene expression levels of Fos-related antigen 2 (FOSL2), peroxisome proliferator-activated receptor gamma (PPARγ), Forkhead Box C1 (FOXC1), and Meis homeobox 2 (Meis2) (Figure 2A,B). Under normal conditions, the mRNA levels of both FOSL2 and FOXC1 were significantly reduced in the cells transfected with the miR-204-5p mimics (*p* = 0.006 and *p* = 0.002, respectively). During LPS-induced inflammation, the mRNA levels of FOXC1 also showed a significant reduction following miR-204-5p mimic transfection (*p* = 0.003) (Figure 2A). FOSL2 and FOXC1 protein levels were also analyzed (Figure 2B). FOSL2 protein levels showed a significant reduction in the miR-204-5p-transfected cells following the LPS treatment (*p* = 0.008). Similarly, the FOXC1 protein levels were significantly decreased in the presence of LPS in the miR-204-5p-transfected cells compared to those in the control transfection group (*p* = 0.020).

### 2.2. The Effects of the miR204-5p Mimics on the mRNA and Protein Levels of Genes Involved in Cell Maintenance and Inflammation

Another aspect of miR-204-5p overexpression is its potential impact on the mRNA expression levels of proteins involved in LEC maintenance and inflammation, such as ATP-binding cassette transporter subfamily G member 2 (ABCG2), Prostaglandin E Synthase 2 (PTGES2), interleukin-1β (IL-1β), and interleukin-6 (IL-6). The mRNA levels of ABCG2, PTGES2, and IL-1β remained unchanged following transfection with the miR-204-5p mimics and/or following LPS-induced inflammation (*p* ≥ 0.051). However, LPS-induced inflammation significantly increased the mRNA levels of IL-6 in the primary LECs in the presence of the miR-204-5p mimics (*p* = 0.002). Additionally, under the inflammatory conditions induced by LPS, the miR-204-5p mimics further upregulated IL-6 mRNA levels (*p* = 0.029). (Figure 3A). The IL-6 protein levels in the cell lysate were measured using ELISA, and no significant differences were observed among the groups (*p* ≥ 0.999) (Figure 3B).

### 2.3. The Effects of the miR204-5p Mimics on the mRNA and Protein Levels of Genes Involved in Maintaining Cell Structure and Matrix Remodeling

Another important group of genes in LECs includes those encoding proteins involved in maintaining cell structure and matrix remodeling. Therefore, we examined the effects of miR-204-5p on the gene expression levels of keratin 3 and 12 (KRT3 and KRT12), as well as Matrix Metalloproteinases 2 and 9 (MMP2 and MMP9), under both normal and LPS-induced inflammatory conditions. Under normal conditions, transfection with the miR-204-5p mimics significantly reduced the KRT12 mRNA levels (*p* < 0.0001). During LPS-induced inflammation, transfection with the miR-204-5p mimics led to a significant downregulation in the mRNA levels of KRT3 and KRT12 in the primary LECs compared to those in the control-transfected cells (*p* = 0.010 and *p* < 0.0001, respectively). Additionally, the LPS treatment further reduced the mRNA levels of KRT3 and MMP2 in the miR-204-5p-transfected cells compared to those in the same group in the absence of LPS (*p* = 0.021 and *p* = 0.004, respectively) (Figure 4A). These mRNA-level changes were assessed at the protein level using a Western blot analysis. Among the tested proteins, only KRT3 showed a significant reduction upon the LPS treatment in the miR-204-5p-transfected cells (*p* = 0.03). The KRT12 and MMP2 protein levels did not show significant differences between groups (*p* ≥ 0.593) (Figure 4B).

### 2.4. The Effects of the miR204-5p Mimics on the mRNA and Protein Levels of Genes Involved in Retinoic Acid Signaling

The effects of miR-204-5p on the expression levels of genes involved in retinoic acid (RA) signaling were analyzed (Figure 5 and Figure 6). RA, a metabolite of vitamin A, regulates several key cellular processes, including differentiation, proliferation, and apoptosis [13]. It also plays a crucial role in maintaining the balance between proliferation and differentiation [14]. We assessed the expression levels of the Retinoic Acid Receptors α and β (RARA and RARB), Retinoid X Receptors α and β (RXRA and RXRB), and retinoic acid-binding proteins, such as Retinol-Binding Protein 1 (RBP1), Cellular Retinoic-Acid-Binding Protein 2 (CRABP2), and Fatty-Acid-Binding Protein 5 (FABP5). Additionally, we examined retinol dehydrogenases, including Retinol Dehydrogenase 10 (RDH10), Alcohol Dehydrogenase 7 (ADH7), and Alcohol Dehydrogenase 1A1 (ADH1A1), as well as enzymes involved in the conversion of retinol into retinaldehyde and RA, such as Cytochrome P450 Family 1 Subfamily B Member 1 (CYP1B1) and Cytochrome P450 26A1 (CYP26A1), in the LECs.

Under normal conditions, transfection with the miR-204-5p mimics led to a significant reduction in the mRNA levels of RXRA and RDH10 (*p* = 0.011, *p* = 0.0007, respectively). During LPS-induced inflammation, the miR-204-5p mimics downregulated the RDH10 and FABP5 mRNA levels in the primary LECs (*p* = 0.039, *p* = 0.001). However, no significant changes were observed in the expression levels of the other mRNAs analyzed or in the protein levels of RXRA, RDH10, and FABP5 in response to the miR-204-5p mimics and/or LPS-induced inflammation (*p* ≥ 0.108) (Figure 5 and Figure 6).

## 3. Discussion

The most conspicuous finding of our study is that miR-204-5p can modulate the mRNA levels of FOXC1, FOSL2, IL-6, KRT3, KRT12, MMP2, RXRα, RDH10, and FABP5, as well as the protein levels of FOXC1and KRT3, during inflammation.

In a previous study, the PAX6 protein levels remained unchanged following the transfection of primary human LECs with miR-204-5p mimics [9]. Therefore, the gene expression changes analyzed in the present study cannot be attributed to the PAX6 knockdown induced by the mimics but rather an independent effect of the miR204-5p mimics.

Previous reports have shown that the expression levels of FOXC1, FOSL2, KRT3, KRT12, and certain components of the retinoic acid signaling pathway, such as ADH7 and ALDH1A1, are downregulated while IL-6 levels may be upregulated in cases of PAX6 haploinsufficiency [12,15,16]. Additionally, other gene expression changes, such as decreased levels of FABP5 and RBP1, have also been described as consequences of PAX6 knockdown in primary LECs [17]. Interestingly, in the present study, we observed that the miR204-5p mimics induced some similar gene expression changes, such as reduced levels of KRT3, KRT12, and FABP5 mRNA. In addition, the KRT3 protein levels decreased following miR-205-5p mimic transfection in combination with the LPS treatment. These findings also indicate that KRT3 and KRT12, early markers of corneal epithelial cell differentiation, are reduced after the use of miR-204-5p mimics and LPS treatment. However, the mRNA levels of ABCG2, which are associated with stem-cell-like properties, remained unchanged, suggesting that LECs reach but do not surpass an undifferentiated state. LEC proliferation is also decreased following the use of miR-204-5p mimics.

We observed significant downregulation of FOXC1 following the overexpression of miR-204-5p under both normal and inflammatory conditions. FOXC1 has been identified as a target of miR-204-5p in endometrioid endometrial cancer cells, where its downregulation—achieved either through siRNA knockdowns or miR-204-5p overexpression—led to a significant reduction in cell migration [18]. Similarly, another study reported decreased miR-204-5p levels and increased FOXC1 expression in laryngeal squamous cell carcinoma (LSCC) compared to those in control cells, correlating with the cancer stage and malignancy. The treatment of these cells with miR-204-5p mimics or FOXC1 siRNA reduced the cell proliferation and colony formation while inhibiting cell migration compared to these properties in controls [19]. The downregulation of FOXC1 by miR-204-5p in these two studies is consistent with our findings. However, the implications of FOXC1 downregulation in aniridia patients warrant further investigation.

Another transcription factor, FOSL2, was also downregulated in the cells transfected with the miR-204-5p mimics and cultured in the normal medium. The FOSL2 gene encodes Fos-related antigen 2 (Fra-2), a member of the transcription factor family involved in cell proliferation and differentiation [20]. FOSL2 is regulated by PAX6, with PAX6 deficiency leading to reduced FOSL2 expression [21]. An RNA sequencing analysis of LECs transfected with FOSL2 siRNA revealed that FOSL2 downregulation results in the upregulation of pro-angiogenic genes, such as JAG1 and TGFB1, in the absence of inflammation [21]. Furthermore, in our recent study on LECs transfected with miR-204-5p, we observed a trend toward a reduced VEGFα mRNA expression in normal medium, with a significant reduction under LPS treatment [9]. These findings collectively suggest that miR-204-5p plays a significant role in regulating neovascularization on the ocular surface. Interestingly, we could not identify any previous studies describing FOSL2 as a novel target of miR-204-5p. Therefore, we propose FOSL2 as a novel target of miR-204-5p in LECs.

In LECs with exogenous expression of miR-204-5p, the IL-6 mRNA expression levels were upregulated in the presence of LPS and were significantly higher compared to those in the control cells. This increase in IL-6 expression is a typical response to LPS-induced inflammation [22]. While miR-204-5p is known to suppress IL-6-induced inflammation by targeting its receptor, IL-6 receptor α (IL6R), no direct effect of miR-204-5p on IL-6 levels has been reported [23]. On the contrary, IL-6 has been proposed to suppress the miR-204-5p expression in oligodendrocytes [24]. Based on these observations, we hypothesize that the elevated IL-6 mRNA expression observed in inflammatory conditions in the miR-204-5p-transfected cells represents a compensatory response to the overexpression of miR-204-5p, aiming to counteract the reduction in IL6R.

Our data also demonstrate a significant reduction in KRT3 and KRT12 mRNA expression levels. These proteins act as structural components in LECs and are well-established markers of differentiated LECs. Reduced expression of KRT3 and KRT12 has also been reported in LECs with aniridia, where it was associated with impaired differentiation [15]. To date, no KRT family members have been identified as direct targets of miR-204-5p. The downregulation of these genes is likely to contribute to reduced cell migration and differentiation. Therefore, our findings suggest that miR-204-5p influences LECs’ structure and differentiation by modulating the expression of KRT3 and KRT12.

The MMP2 expression levels are significantly reduced in cells overexpressing miR-204-5p under LPS-induced inflammatory conditions. Both matrix metalloproteinases (MMP2 and MMP9) play a crucial role in degrading collagen within the extracellular matrix, thereby facilitating cell migration. MMP2 and MMP9 have been implicated in promoting cell migration, viability, and the secretion of angiogenic factors in retinoblastoma cells [25]. Consistent with our findings, the overexpression of miR-204-5p in human corneal epithelial cells has previously been shown to significantly reduce cell proliferation and migration [7]. Thus, the downregulation of MMP2 under inflammatory conditions is likely a contributing factor to the observed reduction in cell migration.

Among the genes involved in retinoic acid signaling, we observed a significant decrease in the RXRA and RDH10 mRNA expression levels following the exogenous expression of miR-204-5p in the normal medium, as well as decreased levels of RDH10 and FABP5 under inflammatory conditions. RXRα plays a key role in modulating the transcription of retinoic acid target genes by forming a heterodimer with RAR and binding to the enhancer regions [26]. RDH10 is essential in the vitamin A metabolic pathway, converting all-trans-retinol into all-trans-retinal, a precursor for retinoic acid, and is critical for embryonic development [27]. A reduction in RDH10 and RXRA mRNA levels would likely disrupt retinoic acid signaling. Previous studies on conjunctival cells from aniridia patients have reported the significant downregulation of miR-204-5p, alongside the upregulation of RDH10 expression [11]. This aligns with our findings, where increased miR-204-5p expression resulted in reduced RDH10 levels.

Furthermore, RA is transported by two carrier proteins, CRABP2 and FABP5, with the balance between these transporters determining a cell’s fate. Higher FABP5 levels promote cell survival and proliferation, while increased CRABP2 levels inhibit cell growth and trigger apoptosis. Thus, the observed reduction in the FABP5 mRNA expression following miR-204-5p overexpression, along with the tendency towards increased CRABP2 levels, is likely to initiate apoptosis. In contrast to our findings, LECs with aniridia exhibit reduced FABP5 levels despite the decreased miR-204-5p expression compared to those in healthy subjects. Notably, PAX6 downregulation leads to decreased FABP5 expression in cells treated with PAX6 siRNA [17]. Therefore, in these patients, reduced miR-204-5p levels cannot elevate FABP5 expression, as it is strongly downregulated due to PAX6 haploinsufficiency. To date, no studies have identified RDH10, RXRA, or FABP5 as direct targets of miR-204-5p.

The effects of the miR-204-5p mimics on genes involved in LEC differentiation, migration, and inflammatory responses under both normal and inflammatory conditions are summarized in Figure 7. The connections and interactions between these genes were investigated using a core data resource for protein–protein interaction networks, the STRING database.

In conclusion, miR-204-5p modulates the expression of a network of genes impacting LEC differentiation, migration, and responses to inflammation. In addition, under inflammatory conditions, miR-204-5p modulates FOSL2 and KRT3 at the protein levels, which highlights these proteins as novel targets.

## 4. Materials and Methods

### 4.1. Isolation and Culture of the Primary Limbal Epithelial Cells

Primary LECs were isolated from corneoscleral rims provided by the Klaus Faber Center for Corneal Diseases, including the Lions Eye Bank, following the use of the corneas for corneal transplantation (the donor information is summarized in Table 1). To isolate the LECs, the limbal area was punched using a 1.5 mm punch (Kai Medical, Solingen, Germany), and the tissue pieces were then incubated overnight in 0.5 mg/mL Collagenase A (Roche Pharma AG, Basel, Switzerland). The next day, the cell suspension was passed through a 20 μm Cell Trics filter (Sysmex Partec GmbH, Gorlitz, Germany), and the cells attached to the filter were washed with 3 mL of trypsin–EDTA solution (Sigma-Aldrich GmbH, Deisenheim, Germany). Then, the cells were incubated for 10 min with trypsin at 37 °C and 5% CO_2_, and afterward, 3 mL of DMEM containing 10% FCS was added to stop the reaction. After centrifugation at 200 g for 5 min, the cells were cultured in KSFM medium (Cat. Nr. C-20111, Promocell, Heidelberg, Germany) at 37 °C and 5% CO_2_.

### 4.2. Transfection and LPS Treatment of the Primary LECs

The primary LECs were transfected using the Lipofectamine 2000 reagent (Cat. Nr. 11668027, Invitrogen, Hercules, CA, USA), following the manufacturer’s instructions. For transfection, 100 pg/μL of hsa-miR-204-5p mimic RNA or the negative control (ctrl) miRNA (Cat. Nr. 4464066 and 4464061, Thermo Fisher Scientific, Waltham, MA, USA) was used. The transfection mixture was applied dropwise to the corresponding well, and the plates were incubated at 37 °C and 5% CO_2_ overnight. Thereafter, the medium was changed, and the cells were collected 72 h after transfection for further analysis. The impact of an inflammatory environment in the cells with miR204-5p overexpression was evaluated by adding 2 μg/mL of lipopolysaccharide (LPS) (Sigma-Aldrich, St. Louis, MO, USA), which was added to the culture medium 48 h after transfection. Cells were collected 72 h after transfection.

### 4.3. The BrdU Assay

The cell proliferation was evaluated using the colorimetric BrdU Cell Proliferation ELISA kit (Merck KGaA, Darmstadt, Germany), following the manufacturer’s protocol. Absorbance readings were obtained using the Tecan Infinite F50 Absorbance Microplate Reader (Tecan Group AG, Männedorf, Switzerland). The data were normalized to the control group, which was set at 100%.

### 4.4. Protein, RNA, and cDNA Extraction

RNA and protein isolation was achieved using an RNA/protein isolation kit (Cat. Nr. 47700, Norgen Biotek CORP., Toronto, ON, Canada). The cells were collected in the lysis buffer provided by the manufacturer (SKP Buffer), and all of the steps were performed following the manufacturer’s protocol. The RNA concentration was measured according to the absorbance at 260 nm using UV/VIS spectrophotometry (Analytik Jena AG, Jena, Germany). The RNA and protein samples were stored at −80 °C until further experiments. The One Taq^®^ RT-PCR Kit (New England Biolabs INC, Frankfurt, Germany) was used for reverse transcription, with 1 μg of RNA as the template.

### 4.5. Quantitative PCR (qPCR)

The quantitative PCR (qPCR) experiments were performed using a SYBR Green kit (Qiagen N.V., Venlo, The Netherlands). For each reaction, 9 µL of reaction buffer was prepared containing 1 µL of the specific primer solution (QuantiTect Primer Assay, Qiagen N.V., Venlo, The Netherlands), 5 µL of SYBR Green Mix, and 3 µL of nuclease-free water. To each reaction, 1 µL of cDNA was added, and samples were then placed into a QuantStudio 5 real-time PCR system (Thermo Fisher Scientific, Waltham, MA, USA). The thermal cycling conditions for amplification were set to 95 °C for 10 s, 60 °C for 30 s, and 95 °C for 15 s, repeated for 40 cycles. The results were analyzed and expressed as the fold change relative to the ctrl miRNA (2^−(Δ ΔCt)^). The primers used for qPCR are listed in Table 2.

### 4.6. The Western Blot Analysis

Alterations in the intracellular protein concentrations were measured using a Western blot analysis. For this purpose, the total protein concentrations were measured using the Bradford assay (Sigma-Aldrich, Merck KGaA, Darmstadt, Germany), using bovine serum albumin (BSA) as the standard. The absorbance was measured at 595 nm using a Tecan Infinite F50 Absorbance Microplate Reader (Tecan Group AG, Männedorf, Switzerland). The protein concentrations were calculated using the linear equation for the BSA standard curve. For electrophoresis, 20 µg of protein was mixed with 4× Laemmli buffer and denatured through heating at 95 °C for 5 min. The samples were then loaded onto NuPAGE ™ Bis-Tris Precast 4–12% gels (Thermo Fisher Scientific™, Waltham, MA, USA) Following protein separation, the proteins were transferred onto a nitrocellulose membrane (BioRad, Hercules, CA, USA) using the Trans Blot Turbo Transfer System (BioRad, Hercules, CA, USA). The blots were incubated overnight at 4 °C with primary antibodies diluted in WesternFroxx anti-rabbit or anti-mouse HRP solution containing blocking reagent and secondary antibodies (BioFroxx GmbH, Einhausen, Germany). The blots were then washed 3 times with WesternFoxx washing solution, and the signal was enhanced using enhanced chemiluminescence (ECL buffers) (PerkinElmer, Waltham, MA, USA). The bands were visualized and analyzed using the iBright imaging system (Thermo Fisher Scientific, Waltham, MA, USA) and the iBright 1500 system (Invitrogen, Hercules, CA, USA). To ensure an equal amount of protein loading, the protein band intensities were normalized to the β-actin protein intensity in each respective lane. The primary antibodies used in this study are listed in Table 3.

### 4.7. The Enzyme-Linked Immunosorbent Assay (ELISA)

To measure the IL-6 concentrations inside the cells, cell lysates were used for protein detection using the ELISA technique. For this purpose, the Human IL-6 DuoSet ELISA kit was purchased from R&D (Minneapolis, MN, USA, DY206), and the experiments were conducted following the manufacturer’s protocol. First, the capture antibody was coated onto each well of the plate and left to incubate overnight at room temperature. Next, 100 μL of the collected supernatant was added to each well and incubated for 2 h. The detection antibody was then applied and allowed to incubate for an additional 2 h to facilitate the formation of an antibody–analyte–detection antibody sandwich complex.

For quantification, a calibration curve was generated using human recombinant proteins as the standards. The optical density (OD) of each well was measured using the Tecan Infinite F50 Absorbance Microplate Reader (Tecan Group AG, Männedorf, Switzerland). To normalize the results, the OD values were divided by the total protein concentration of the corresponding flask, expressed as picograms (pg) per milligram of protein (mg). These normalized values were then used for further statistical analysis.

### 4.8. Statistical Analysis

The statistical analysis was performed using GraphPad Prism software (Version 10.2.1). The data are shown as means ± SEMs, and significance was assessed using a two-way ANOVA followed by Bonferroni’s correction. Values were normalized to the respective controls before comparison. *p*-values below 0.05 were considered statistically significant.

## Figures and Tables

**Figure 1 ijms-26-03809-f001:**
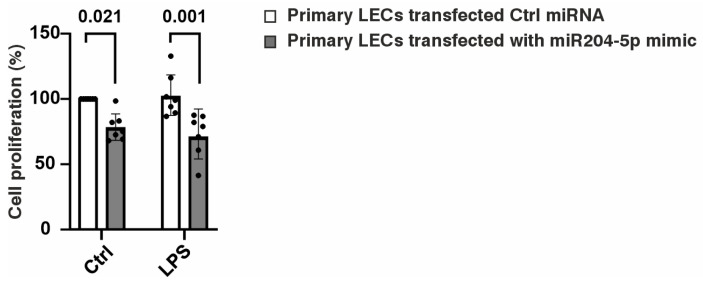
Effect of miR204-5p mimics on limbal epithelial cell (LEC) proliferation. The BrdU assay demonstrated a significant reduction in the LEC proliferation following treatment with the miR-204-5p mimics, both in the untreated and LPS-treated LECs (*p* = 0.021, *p* = 0.001). However, LPS itself had no impact on LEC proliferation. Each dot represents an individual donor.

**Figure 2 ijms-26-03809-f002:**
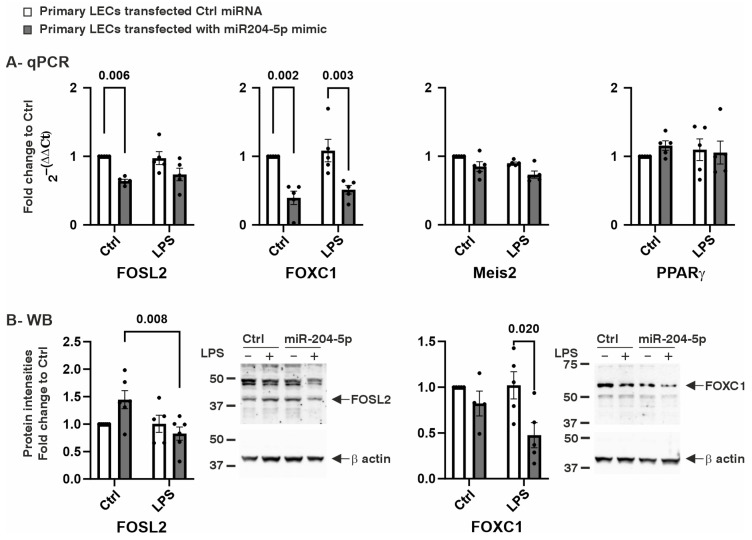
Effect of miR204-5p mimics on mRNA and protein levels of transcription factors Fos-like antigen 2 (FOSL2) and Forkhead Box C1 (FOXC1), homeobox transcription factor Myeloid Ectopic Viral Integration Site 2 (Meis2), and adipocyte differentiation regulator peroxisome proliferator-activated receptor gamma (PPARγ) in primary human limbal epithelial cells (LECs) using normal medium (Ctrl) or medium containing lipopolysaccharide (LPS). Data are represented as means ± SEMs. The statistical analysis was performed using a two-way ANOVA, followed by Bonferroni’s test. *p*-values below 0.05 were considered statistically significant. (**A**) FOSL2 and FOXC1 mRNA levels were downregulated in the cells transfected with the miR204-5p mimics in normal medium (*p* = 0.006, *p* = 0.002). Under LPS-induced inflammation, mRNA levels of FOXC1 were downregulated following transfection with the miR204-5p mimics (*p* = 0.003). Nevertheless, none of the levels of the other mRNAs examined changed significantly upon the use of the miR204-5p mimics and/or during LPS-induced inflammation (*p* ≥ 0.07). (**B**) FOSL2 protein levels were significantly reduced in miR-204-5p-transfected cells upon the treatment with LPS (*p* = 0.008). FOXC1 protein levels were significantly reduced in the presence of LPS in the miR204-5p-transfected cells compared to those in control transfection (*p* = 0.020). Each dot represents an individual donor.

**Figure 3 ijms-26-03809-f003:**
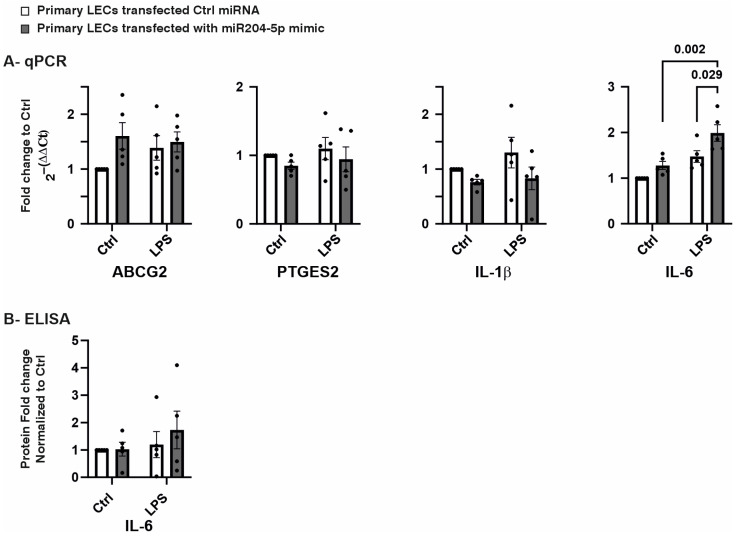
Effects of miR204-5p mimics on mRNA and protein levels of genes involved in cell maintenance and inflammation in primary human limbal epithelial cells (LECs) using normal medium (Ctrl) or medium containing lipopolysaccharide (LPS) (**A**,**B**). Data are represented as means ± SEMs. The statistical analysis was performed using a two-way ANOVA, followed by Bonferroni’s test. *p*-values below 0.05 were considered statistically significant. (**A**) ATP-Binding Cassette Subfamily G Member 2 (ABCG2), Prostaglandin E Synthase 2 (PTGES2), and interleukin 1 β (IL-1β) mRNA levels did not change significantly upon the use of the miR204-5p mimics and/or LPS-induced inflammation (*p* ≥ 0.051). Nevertheless, LPS-induced inflammation upregulated the mRNA levels of interleukin-6 (IL-6) in the primary LECs transfected with the miR204-5p mimics (*p* = 0.002). In addition, under LPS-induced inflammation, the miR204-5p mimics significantly increased IL-6 mRNA levels (*p* = 0.029). (**B**) The IL-6 protein levels in the cell lysate did not change significantly upon the use of the miR204-5p mimics and/or LPS-induced inflammation (*p* ≥ 0.999). Each dot represents an individual donor.

**Figure 4 ijms-26-03809-f004:**
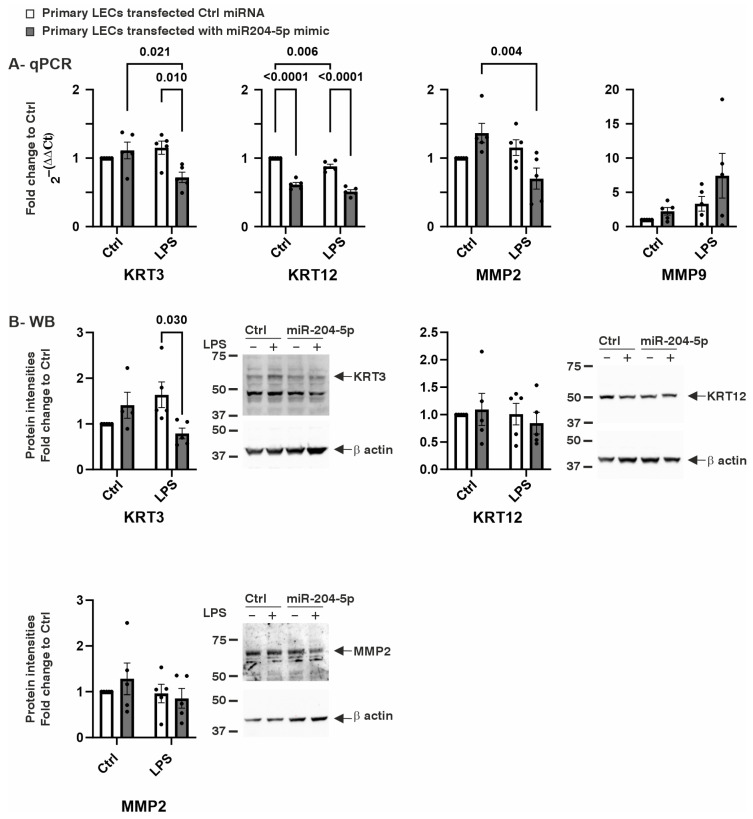
Effect of miR204-5p mimics on mRNA and protein levels of structural proteins keratin 3 (KRT3) and keratin 12 (KRT12) and matrix remodeling proteins such as Matrix Metallopeptidase 2 (MMP2) and Matrix Metallopeptidase 9 (MMP9) in primary human limbal epithelial cells (LECs) using normal medium (Ctrl) or medium containing lipopolysaccharide (LPS). Data are represented as means ± SEMs. The statistical analysis was performed using a two-way ANOVA, followed by Bonferroni’s test. *p*-values below 0.05 were considered statistically significant. (**A**) In the normal medium, the overexpression of miR204-5p led to the downregulation of KRT12 mRNA levels (*p* = 0.006). Under LPS-induced inflammation, the mRNA levels of KRT3 and KRT12 were downregulated in the primary LECs following the use of the miR204-5p mimics (*p* = 0.010, *p* < 0.0001). Furthermore, in the cells transfected with miR204-5p, LPS-induced inflammation resulted in reduced KRT3 and MMP2 mRNA levels (*p* = 0.021, *p* = 0.004). Nevertheless, none of the other mRNA levels examined changed significantly upon the use of the miR204-5p mimics and/or LPS-induced inflammation (*p* ≥ 0.06). (**B**) Under LPS-induced inflammation, the KRT3 protein levels were significantly downregulated following the use of the miR204-5p mimics (*p* = 0.030). The KRT12 and MMP2 protein levels did not differ significantly between groups (*p* ≥ 0.593). Each dot represents an individual donor.

**Figure 5 ijms-26-03809-f005:**
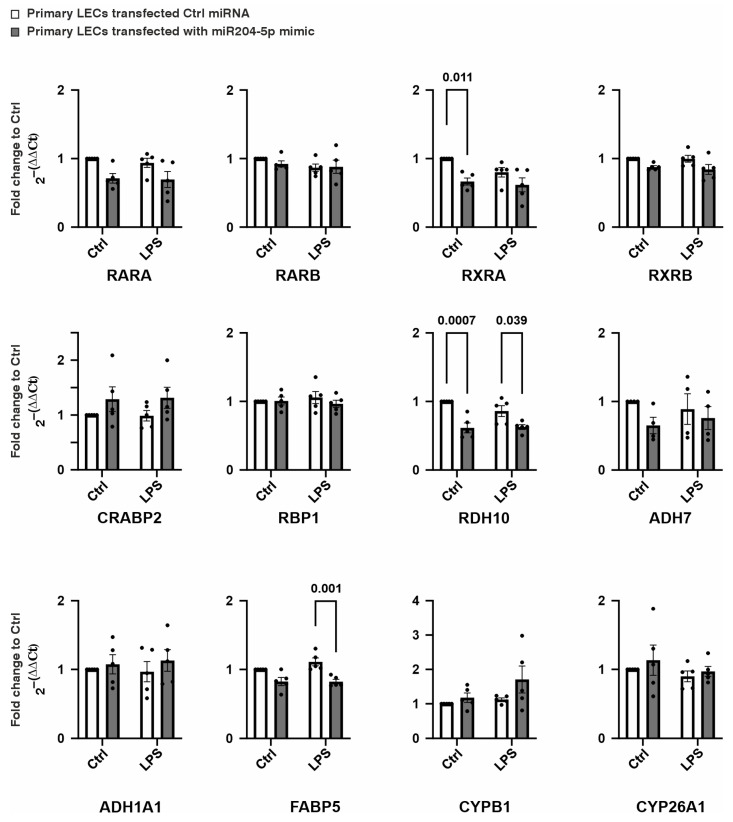
The effect of the miR204-5p mimics on the mRNA levels of genes involved in retinoic acid signaling in primary human limbal epithelial cells (LECs) using normal medium (Ctrl) or medium containing lipopolysaccharide (LPS). Data are represented as means ± SEMs. The statistical analysis was performed using a two-way ANOVA, followed by Bonferroni’s test. *p*-values below 0.05 were considered statistically significant. In the normal medium, treating the cells with the miR204-5p mimics downregulated the mRNA levels of Retinoid X Receptor Alpha (RXRA) and Retinol Dehydrogenase 10 (RDH10) (*p* = 0.011, *p* = 0.0007). Under LPS-induced inflammation, the mRNA levels of RDH10 and Fatty-Acid-Binding Protein 5 (FABP5) were downregulated in the primary LECs following the use of the miR204-5p mimics (*p* = 0.039, *p* = 0.001). Nevertheless, none of the other mRNA levels examined changed significantly upon the use of the miR204-5p mimics and/or LPS-induced inflammation (*p* ≥ 0.06). RARA: Retinoic Acid Receptor Alpha; RARB: Retinoic Acid Receptor Beta; RXRB: Retinoid X Receptor Beta; CRABP2: Cellular Retinoic-Acid-Binding Protein 2; RPB1: Retinol-Binding Protein 1; ADH7: Alcohol Dehydrogenase 7; ADH1A1: Alcohol Dehydrogenase 1A1; CYPB1: Cytochrome P450 Family 1 Subfamily B Member 1; CYP26A1: Cytochrome P450 Family 26 Subfamily A Member 1. Each dot represents an individual donor.

**Figure 6 ijms-26-03809-f006:**
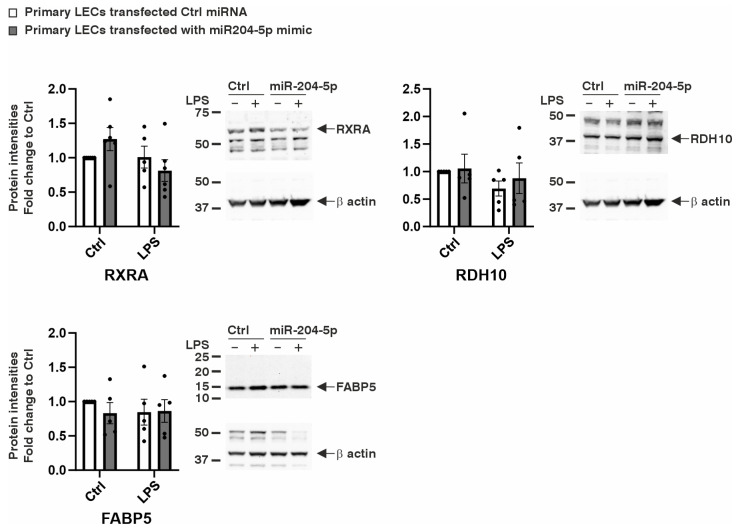
The effect of the miR204-5p mimics on the protein levels of affected genes involved in retinoic acid signaling in primary human limbal epithelial cells (LECs) using normal medium (Ctrl) or medium containing lipopolysaccharide (LPS). Data are presented as means ± SEMs. The statistical analysis was performed using a two-way ANOVA, followed by Bonferroni’s test. *p*-values below 0.05 were considered statistically significant. There were no significant differences in the Retinoid X Receptor Alpha (RXRA), Retinol Dehydrogenase 10 (RDH10), or Fatty Acid-Binding Protein 5 (FABP5) protein levels between any of the groups (*p* ≥ 0.108). Each dot represents an individual donor.

**Figure 7 ijms-26-03809-f007:**
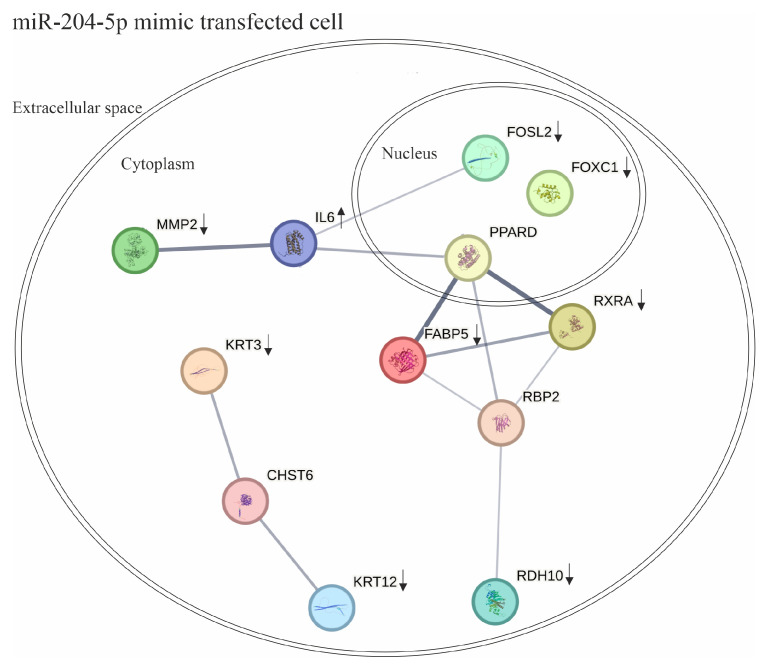
The effects of miR-204-5p mimics on genes involved in limbal epithelial cell (LEC) differentiation, migration, and inflammatory responses under both normal and inflammatory conditions are summarized. The connections and interactions between these genes were explored using the STRING database. CHST6: carbohydrate sulfotransferase 6; FABP5: Fatty-Acid-Binding Protein 5; FOXC1: Forkhead Box C1; FOSL2: Fos-like antigen 2; IL-6: interleukin-6; KRT12: keratin 12; KRT3: Keratin 3; MMP2: Matrix Metallopeptidase 2; PPARD: peroxisome proliferator-activated receptor delta; RXRA: Retinoid X Receptor Alpha; RBP2: Retinol-Binding Protein 2; RDH10: Retinol Dehydrogenase 10. Each color represents a protein and arrows represent the changes that we observed upon overexpression of miR-204-5p (upwards: upregulation and downwards: down regulation).

**Table 1 ijms-26-03809-t001:** Age and gender of the donors included in this study. n/a: not available.

Donor Nr.	Age (Years)	Gender
Donor 1	85	Female
Donor 2	n/a	Male
Donor 3	73	Female
Donor 4	n/a	Male
Donor 5	n/a	Male
Donor 6	n/a	Female
Donor 7	n/a	Male

**Table 2 ijms-26-03809-t002:** Primers used for qPCR analysis.

Primer	Gene Globe ID	Manufacturer
Hs_RARA_1_SG QuantiTect Primer Assay	QT00095865	Qiagen N.V., Venlo, The Netherlands
Hs_RARB_1_SG QuantiTect Primer Assay	QT00062741	Qiagen N.V., Venlo, The Netherlands
Hs_RXRA_1_SG QuantiTect Primer Assay	QT00005726	Qiagen N.V., Venlo, The Netherlands
Hs_RXRB_1_SG QuantiTect Primer Assay	QT00061117	Qiagen N.V., Venlo, The Netherlands
Hs_CRABP2_1_SG QuantiTect Primer Assay	QT00063434	Qiagen N.V., Venlo, The Netherlands
Hs_RBP1_2_SG QuantiTect Primer Assay	QT01850296	Qiagen N.V., Venlo, The Netherlands
Hs_RDH10_1_SG QuantiTect Primer Assay	QT00029176	Qiagen N.V., Venlo, The Netherlands
Hs_ADH7_1_SG QuantiTect Primer Assay	QT00000217	Qiagen N.V., Venlo, The Netherlands
Hs_ALDH1A1_1_SG QuantiTect Primer Assay	QT00013286	Qiagen N.V., Venlo, The Netherlands
Hs_FABP5_1_SG QuantiTect Primer Assay	QT0022556	Qiagen N.V., Venlo, The Netherlands
Hs_CYP1B1_1_SG QuantiTect Primer Assay	QT00209496	Qiagen N.V., Venlo, The Netherlands
Hs_CYP26A1_1_SG QuantiTect Primer Assay	QT00026817	Qiagen N.V., Venlo, The Netherlands
Hs_ABCG2_1_SG QuantiTect Primer Assay	QT00073206	Qiagen N.V., Venlo, The Netherlands
Hs_PTGES2_1_SG QuantiTect Primer Assay	QT00082068	Qiagen N.V., Venlo, The Netherlands
Hs_IL1B_1_SG QuantiTect Primer Assay	QT00021385	Qiagen N.V., Venlo, The Netherlands
Hs_IL16_1_SG QuantiTect Primer Assay	QT00075138	Qiagen N.V., Venlo, The Netherlands
Hs_FOSL2_1_SG QuantiTect Primer Assay	QT01000881	Qiagen N.V., Venlo, The Netherlands
Hs_PPARG_1_SG QuantiTect Primer Assay	QT00029841	Qiagen N.V., Venlo, The Netherlands
Hs_FOXC1_1_SG QuantiTect Primer Assay	QT00217161	Qiagen N.V., Venlo, The Netherlands
Hs_MEIS2_1_SG QuantiTect Primer Assay	QT00077315	Qiagen N.V., Venlo, The Netherlands
Hs_KRT3_1_SG QuantiTect Primer Assay	QT00050365	Qiagen N.V., Venlo, The Netherlands
Hs_KRT12_1_SG QuantiTect Primer Assay	QT00011949	Qiagen N.V., Venlo, The Netherlands
Hs_MMP2_vb.1_SG QuantiTect Primer Assay	QT02395778	Qiagen N.V., Venlo, The Netherlands
Hs_MMP9_1_SG QuantiTect Primer Assay	QT00040040	Qiagen N.V., Venlo, The Netherlands
Hs_TBP_1_SG QuantiTect Primer Assay	QT00000721	Qiagen N.V., Venlo, The Netherlands

**Table 3 ijms-26-03809-t003:** Primary antibodies used for Western blot analysis.

Antibody	Class	Dilution	Cat. No.	Manufacturer
RXRa	Polyclonal	1:2000	21218-1-AP	Proteintech, Planegg-Martinsried, Germany
RDH10	Polyclonal	1:1000	14644-1-AP	Proteintech, Planegg-Martinsried, Germany
FABP5	Polyclonal	1:1000	12348-1-AP	Proteintech, Planegg-Martinsried, Germany
FOSL2	Monoclonal	1:3000	TA809660S	OriGene Technologies, Rockville, MD, USA
FOXC1	Polyclonal	1:500	55365-1-AP	Proteintech, Planegg-Martinsried, Germany
Keratin K3/K76	Monoclonal	1:500	CBL218	MERCK, Darmstadt, Germany
Keratin 12 (E-8)	Monoclonal	1:2000	sc-515882	Santa Cruz biotechnology, Dallas, TX, USA
MMP2	Monoclonal	1:250	66366-1-Ig	Proteintech, Planegg-Martinsried, Germany
β-actin	Polyclonal	1:10,000	Ab8227	Abcam, Cambridge, UK

## Data Availability

Dataset available on request from the authors.

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
