# Peer review of "Effect of MiRNA 204-5P Mimics and Lipopolysaccharide-Induced Inflammation on Transcription Factor Levels, Cell Maintenance, and Retinoic Acid Signaling in Primary Limbal Epithelial Cells"

_ijms, 2025, doi:10.3390/ijms26083809_

Round 1
Reviewer 1 Report
Comments and Suggestions for Authors
The authors characterized the functions of MicroRNA miR-204-5p in the primary limbal epithelial cells by characterizing the transcript and protein levels of a few of genes that function as transcription factor, cytokines, or extracellular matrix under normal and LPS-induced inflammation. I can’t be positive for publication of this manuscript.
1. This study is not driven by a specific scientific question or a hypothesis.
2. The manuscript is not well organized. The results-section even has no subtitles to summarize the main results obtained in this study.
3. For Western blot assay, the specificities of the antibodies are low. The binds for the target proteins and internal reference are not derived from the same membrane.
4. The authors haven’t described why they chose to analyze the genes and proteins.
Reviewer 2 Report
Comments and Suggestions for Authors
This manuscript describes the role of microRNA-204-5p in regulating gene expressions related to transcription levels, cell maintenance/inflammation, cell structures, and retinoic acid signaling, using qPCR under both normal and inflammatory conditions. The protein level change was evaluated by ELISA or Western Blot. As a conclusion, the authors proposed new targets of microRNA-204-5p and a model of protein-protein interactions. Overall, this manuscript is appropriate for publishing in International Journal of Molecular Sciences, because it intends to reveal critical molecular regulations of miRNA in a disease-relevant context. However, there are critical issues in their experimental designs and result discussion. Therefore, I recommend this paper be reevaluated after major revision. These issues are listed as follows:
Experimental design:
- The authors only did protein analysis for genes that showed a change in their mRNA level. However, it is known that the change in protein and mRNA is not directly correlated and no difference in mRNA can not ensure that there is no difference in protein synthesis. I recommend the authors evaluate both mRNA and protein for candidate targets in parallel before drawing a conclusion.
- Due to the fact that mRNA change didn’t agree with the protein change, to validate the role of miRNA in controlling cell behaviors, cell proliferation/migration/differentiation should be evaluated using complementary assays (e.g. cell growth rate measurement, cell migration speed measurement, cell differentiation markers).
- After transfection, the level of microRNA-204-5p for control and experimental groups needs to be tested to validate the model.
Discussion:
- In line 224 discussing the change of FOXC1, the hypothesis “ reduced miR-204-5p levels may decrease FOXC1 expression, potentially leading to reduced cell proliferation and increased migration” is confusing. This is contradictory to their result, that miR-204-5p overexpression decreased FOXC1 expression. Also, does miR-204-5p cause an opposite effect on cell migration and proliferation? Why there is reduced proliferation concurrent with increased migration? Please read carefully and correct your statement if there is any misleading information.
- In line 235, the authors cited their previous work that miR-204-5p overexpression, which reduced FOSL2, further decreased VEGF expression. But they also cited a previous study showing FOSL2 reduction led to upregulation of proangiogenic genes, which is contradictory to their finding. They need to clarify the effect of miR-204-5p on vascularization.
- At the end, they proposed FOXC1 and KRT3 as new targets for mi-204-5p, while FOXC1 has been found as a target in cancer cells. So they can not claim it as a new target. In contrast, they proposed FOSL2 as a new target when discussing it. Please clarify “new targets” in conclusion.
Round 2
Reviewer 1 Report
Comments and Suggestions for Authors
The authors have addressed some of my concerns. The manuscript should be thoroughly checked to enhance readability.
